# Costs of Healthcare for Children with Inflammatory Bowel Diseases (IBD) in Poland

**DOI:** 10.3390/children10071112

**Published:** 2023-06-26

**Authors:** Grażyna Markiewicz-Łoskot, Wojciech Chlebowczyk, Tomasz Holecki

**Affiliations:** 1Department of Nursing and Social Medical Problems, School of Health Sciences in Katowice, Medical University of Silesia in Katowice, Medyków 12 Str., 40-751 Katowice, Poland; gloskot@sum.edu.pl (G.M.-Ł.); wchlebowczyk@sum.edu.pl (W.C.); 2Department of Health Economics and Health Management, School of Public Health in Bytom, Medical University of Silesia in Katowice, Piekarska 18 Str., 41-902 Bytom, Poland

**Keywords:** Crohn’s disease, ulcerative colitis, inflammatory bowel disease, children, costs of treatment

## Abstract

The last two decades have seen an increase in the incidence of inflammatory bowel disease (IBD) in many regions of the world, which has had a significant impact on both the social and economic burden of governments and healthcare systems. The aim of this study was to determine the level of hospitalization and outpatient treatment costs for children and adolescents with Crohn’s disease and ulcerative colitis, depending on age, location, and activity of the disease. Methods were a retrospective analysis of the medical documentation of 240 children with IBD, hospitalized in the Gastroenterology Ward, Department of Pediatrics Medical University of Silesia (Katowice, Poland), along the three years follow up. The costs of treatment consisted of calculations of the supply of oral and intravenous drugs, calculations of the costs of laboratory tests, imaging, and consultations, as well as person-day costs. The most important results, determined with high costs of IBD treatment, are associated with younger age, high disease activity, localization in the small intestine in Crohn’s disease (CD), and inflammatory changes in the entire colon in Ulcerative Colitis (UC). During the observation, it was noticed that the shortening of the hospitalization time did not significantly affect the total costs, which remained at a stable level.

## 1. Introduction

In 2017, there were 6–8 million cases of inflammatory bowel disease (IBD) globally, diagnosed in over one million citizens in the United States and two and a half million people in Europe, which is a significant cost for the healthcare system. Additional social costs related to IBD concerning the professional development of patients and deteriorating quality of life of patients should also be remembered. The majority of cases are diagnosed in young people, and their prevalence is constantly increasing. Therefore, the impact of IBD on healthcare systems is increasing exponentially [1,2,3]. The incidence of IBD in pediatric-onset IBD is also increasing, particularly in developing countries. This may be connected with the environmental risk factors associated with the Westernization of society [4,5].

In Poland, the increasing competitiveness in the market of health services and contracting them by the National Health Fund has influenced the interest of the management in financing the operations of hospitals. Not only is the global knowledge of costs incurred by hospitals important, but also the individual elements of the “chain” of costs, which consequently determine the outlays for the treatments of patients. Attention was also paid to sources of information for the costs of hospital operations and their records in financial and accounting systems [6].

Pharmacoeconomics, dealing with the economic evaluation of pharmacotherapy, is of great importance in the treatment of many diseases and is based on the study of the relationship between pharmacotherapy costs and obtained health effects, treatment efficacy, and the impact on the length and quality of life. The goal of treatment of IBD is to achieve long-term remission and prevent recurrence of the disease [7].

Due to different clinical pictures and specificities of the disease associated with a young age, the treatment of pediatric patients with IBD is characterized by therapeutic differences. In children, a more severe course of the disease is observed with a higher percentage of complications, and a higher need for surgical intervention compared to adults, and a more extensive location of lesions. In addition, in pediatric patients, growth and puberty disorders occur [8].

The aim of this study was to determine the level of hospitalization and outpatient treatment costs for children and adolescents with CD and UC, depending on age, location, and activity of the disease.

## 2. Materials and Methods

The study performed a retrospective analysis of the medical documentation of children with IBD hospitalized in the Gastroenterology Ward, Department of Pediatrics, Medical University of Silesia in Katowice (Poland) for 2014–2016. The study included the medical records of all children hospitalized with diagnosed Crohn’s disease and children with ulcerative colitis, respectively, 106 and 134 children. In the analyzed period of time, the children included in the study did not require surgery.

The diagnosis of CD and UC was based on the Porto criteria (2005) and modified Porto criteria (2014) [8,9].

When analyzing the costs of hospitalization of patients, the costs of treatment were taken into account, consisting of the supply of drugs (intravenous and oral), treatments, laboratory tests, imaging, and medical consultations, as well as the cost of stay (hotel night). The analysis was carried out in accordance with the settlement system of the National Health Fund.

Quantitative variables referring to the costs of treating patients for the needs of statistical analyses were transcribed into qualitative variables considering the minimum, maximum, and average values. A study of the non-hospital costs among parents and carers of children with IBD was conducted during the stay of patients in the Department of Gastroenterology and during control visits at the Gastroenterological Outpatient Clinic. We asked 103 parents and caregivers to take part in the study, and 61 of them agreed to the study and completed the survey. The original questionnaire was addressed to the parents and caregivers of treated children.

For statistical analysis of the collected material, parametric and non-parametric dependency tests (Pearson and R Spearman X2) and variable independence tests (V-squared test, X2 test with Yates correction) were used. In addition, the correlation strength between statistically significant variables was examined using appropriate correlation coefficients. Statistical analysis of the collected material was made using the following programs: Microsoft Office 2008 (MS Word, MS Excel, and MS Access) and Statistica 12, with the addition of the Medical Kit.

The exclusion criteria concerned children in the questionnaire survey if the parents did not agree to its conduct. In the case of statistical data, all cases in the period were examined.

Before the collection of data, the consent of the Bioethical Commission of the Medical University of Silesia no. KNW/0022/KB/308/15 was obtained.

## 3. Results

Among 240 children with IBD (628 hospitalizations), there were 106 patients with CD (308 hospitalizations) and 134 patients (320 hospitalizations) with UC. During the study period 2014–2016, there were 628 hospitalizations for IBD. There were 308 hospitalizations for CD, which included 106 patients from 8 to 18 years, with an average age of 14.7 years (100 girls and 208 boys). There were 320 hospitalizations for UC, which included 134 patients aged 1 to 18 years with a mean age of 13.2 years (144 girls and 176 boys). The average age at the time of CD diagnosis was 12.7 years and 11.2 years in UC. The duration of hospitalization of children with CD in the Gastroenterology Ward ranged from 1 to 38 days (the average was 3.9 days), and the time of hospitalization of children with UC ranged from 1 to 34 days (the average was 5 days).

During the three-year follow-up, there was a downward trend in the average drug costs (25%), while there was a systematic increase in costs borne by the department as part of diagnostic tests (15%). In the studied period, a systematic decrease in the total costs of hospitalization of patients with CD and in children with UC on conventional treatment was found. However, there was no significant difference. The total average costs included direct costs of tests and medicines, as well as the costs of children staying in the ward, totaling 628,57 EUR. Disease type, i.e., Crohn’s disease and UC, had a statistically significant impact on the incurred costs. Higher costs (727.40 EUR) of treatment (drug costs, research costs, and total costs of hospitalization) were observed in children treated for CD, compared to UC (*p* < 0.04).

We also analyzed the costs of biological treatment of 9 children with Crohn disease aged 11–16 years. The cost of biological treatment (Infliximab) depends on the child’s weight (5 mg/kg bw/dose). Depending on body weight, hospitalization costs ranged from EUR 672.78 to EUR 987.32 (one dose of biologics—EUR 450.74–782.98). Annual biological treatment ranged from EUR 5362.72 to 7458.25 (including biological agents 3428–6408 EUR).

A statistically significant relationship was found between the age of children with CD and the costs of hospitalization (*p* < 0.01). The highest costs were recorded in patients aged up to 9 years, while in cases of children aged 10–14 years, the costs were 38% lower. Differences in hospitalization costs depending on age were also found among children with UC (*p* < 0.01) (Table 1).

The number of hospitalized patients did not differ significantly depending on the disease location. In the examined group of children with CD, the location in the small intestine and ileocecal region dominated, compared to the colon. In the examined group of patients with UC, the least frequent were hospitalized children with rectal inflammation, slightly more often with lesions in the left large intestine, and most often, with inflammation of the entire large intestine. In both CD and UC, a statistically significant relationship was found between the location of the disease and the costs of hospitalization (*p* < 0.01). The highest costs were found in children with CD, where the disease was located in the small intestine and ileocecal region, and in children with UC with inflammation of the entire large intestine (Table 2).

In both diseases, a statistically significant relationship was observed between the activity of the disease and the costs of hospitalization (*p* < 0.01). The highest costs were recorded among children with severe disease (Table 3).

Except for hospital costs incurred by the public payer (National Health Fund), we conducted a survey among 61 parents and caregivers. The studied group of children consisted of 35 boys and 26 girls aged from 3.5 to 18 years. It was found that 59% (36 children) of the surveyed children were treated for UC and 41% (25 children) for CD. The average age of the surveyed children was 14.4 years and did not differ significantly depending on the form of the disease. The duration of the disease also did not show differences due to the form of the disease, and it was approximately 3 years. In more than half (53%) of children with UC, the inflammation involved the whole large intestine. Among patients with CD, the small intestine was mainly occupied (48%). The costs incurred by parents and carers of children with IBD include dietary costs, travel to treatment costs, and the costs of drugs taken on a permanent basis and periodically. The average monthly costs borne by parents and caregivers with UC amounted to 146.98 EUR and were lower by about 20% than the costs borne by parents and carers of children with CD, worth 182.56 EUR (Table 4).

54.1% (33 persons) of surveyed parents thought that their incomes were insufficient and did not allow for the purchase of essential drugs and to meet other expenses associated with the child’s disease. The same percentage of parents (54.1%, 33 persons) used the childcare allowance.

## 4. Discussion

The incidence of IBD continues to rise, and so does the cost. The cost of caring for IBD has almost doubled over the past two decades [10]. The majority of the analyses of the cost of IBD treatment available in the literature were conducted for adult patients. Different methods of calculation were also used, e.g., statistical methods, annual treatment costs, and including or excluding indirect costs, conservative, and surgical treatments. However, comparisons with other studies may be useful to illustrate changes in the cost of healthcare for this group of patients. Data regarding the cost of treatment for children are scarce [11,12,13].

In 2020, Matary conducted a systematic review of the literature on the treatment of IBD in children, including only nine studies, most of which describe the direct costs of treatment in 2005–2016 [14]. The form of the disease has a statistically significant effect on the amount of incurred costs. Higher medical expenses (drug costs, research costs, and total costs of hospitalization) were observed in children treated for CD. Fast and accurate diagnosis of IBD, particularly in children, may reduce the number of hospitalizations and shorten the duration. In the analyzed period, a downward trend in the average drug costs was observed in the conventional treatment, while there was a systematic increase in the costs incurred by the department as part of diagnostic tests.

Prenzler, A. et al. in Germany analyzed the total costs of treating 511 adult patients with CD and determined them on the average at 3767 EUR, of which only 134 patients required hospitalization (26%). The costs of hospitalization were dependent on the disease activity and, on average, ranged from approximately 650 EUR to over 1000 EUR. Scientists from Germany showed approximately 30% higher costs of treatment in cases of severe CD form (Crohn’s Disease Activity Index > 220) while being similar in remission to mild to moderate forms. In UC, the costs of treatment of mild to moderate forms were twice as high as in remission, and the costs were three times higher in severe form. Moreover, Cohen, R. showed three times higher costs, both indirect and direct, in patients with moderate to severe UC and a higher frequency of hospitalizations [15,16].

A. Basi in the United Kingdom showed that patients’ hospitalizations accounted for half of all costs. Similar conclusions from the research were drawn by Longobardi, T. in Canada and Heaton, P.C. in the United States [17,18,19]. There was a relationship between the location and activity of the disease and the age of patients at the time of diagnosis. In the younger age group, under 14 years of age, the disease was most often located in the ileocecal area. In contrast, the most severe forms of CD were observed in children aged 15–18 years. There were no statistically significant correlations between the costs of treating UC and the patients’ age in the study. In the majority of patients (59%), the disease was located throughout the large intestine, regardless of age, whereas the location in the left large intestine and rectum was more common in older age groups.

Kappelman et al. analyzed the direct costs of healthcare for patients with IBD in the United States, both children and adults. In the analyzed group, of almost 20,000 patients, 8% were children, both in the CD and UC group. The average cost of hospitalization for a patient with CD was 1567 USD and was 30% higher than the cost of treatment of UC, which was 1099 USD. The treatment costs were significantly higher in children and adolescents under 20 years of age, compared to adults, but did not show a difference in relation to gender. Higher costs in children were caused by a more severe course of the disease, which resulted in more frequent controls in the specialist clinic and a higher frequency of hospitalization, which was confirmed by the analysis of our research. Bickstone also compared the costs of treatment of UC in three age groups, including those under 18, stating almost twice as high costs of treatment in the group of children [12,20]. The highest costs were found among patients in whom PCDAI was classified as severe. In UC, the relationship between the disease activity and costs in each of the analyzed cases was statistically significant. In the available literature, there are few studies on the relationship of costs related to disease activity in children.

Van der Valk, M. et al., during the three-year observation, also showed the relationship between activity and incurred costs. The most common locations of CD were in the small and large intestine at 50% (in the small intestine at 20% and in the large at 28%). Unfortunately, they did not analyze the costs related to the location of the disease [21,22].

Gibson, P. in Australia showed double costs of treating moderate and severe UC over three months of treatment. It appears that the location of the disease is also related to the severity, and the costs can be compared to the activity of the disease [23].

There are many studies assessing the quality of life of IBD patients, while few relate to the costs of outpatient care in chronically ill children with IBD [24,25].

Sin, T. analyzed the questionnaires of 67 children with CD and 83 with UC, where the mean age of patients was 14 years, and the mean duration of the disease was 3.7 years. The annual costs incurred by parents and caregivers were above 500 USD for 63.5%, above 1000 USD for 28.5%, and above 5000 USD for 5% of patients [26].

Stark, R. conducted cost analysis among 480 adult patients with IBD, and found the average monthly cost was 1425 EUR for CD and was significantly higher than for patients with UC, which was 1015 EUR. Of the total costs of CD treatment, 64% were indirect costs related to early retirement or sick leave, and only 32% were direct costs related to treatment. In the case of UC, 54% of the total costs were indirect, and 41% were direct medical costs [27].

Among surveyed parents, 54.1% believed that their income was insufficient and did not allow them to buy necessary drugs and meet other expenses associated with the disease of their child. In addition, the same percentage of parents (54.1%) benefit from childcare allowance. In Poland, the amount of care allowance in connection with a child’s illness is 35.34 EUR per month. However, the demonstrated costs exceed this amount by several times. Therefore, support in this respect should likely be higher.

It should be noted that the strength of the work is a multifaceted collective analysis of both direct and indirect costs of IBD treatment in children, currently available in a few publications, and the limitations of the study are the small number of children suffering from the discussed diseases and the limited number of publications in the field of health economics.

## 5. Conclusions

The chronic nature of IBD has a large impact on the costs of hospital and outpatient treatment.

In the study group of children, high costs of IBD treatment were associated with high disease activity, localization in the small intestine in CD, inflammatory changes in the entire colon in UC, and younger ages of sick children.During the three-years’ time of observation, we noted a shortening time of hospitalization, decreased costs of treatment, and increased costs of diagnostics tests, in particular, in children with CD, while the total cost of hospitalization remained stable.The costs of outpatient treatment of the examined patients borne by parents and caregivers are high in both types of the disease, without the possibility of meeting all the needs related to the disease of children in half of the surveyed families, indicating the need for financial support from social services.

## Figures and Tables

**Table 1 children-10-01112-t001:** The dependence of hospitalization costs for the studied group of children by age (EUR).

	Crohn Disease		Ulcerative Collitis	
Age Group	N	Cost of Hospitalization	N	Cost of Hospitalization
0–4 years	-	-	19	574.94 ± 556.53
5–9 years	9	1095.82 ± 721.49	50	748.44 ± 692.80
10–14 years	88	672.79 ± 453.12	80	589.14 ± 508.33
15–18 years	211	731.02 ± 519.34	171	617.60 ± 477.33
Without division according to age groups	308	725.02 ± 510.93	320	628.40 ± 528.68

**Table 2 children-10-01112-t002:** The dependence of hospitalization costs for the studied group of children on the location of the disease (EUR).

Crohn Disease			Ulcerative Collitis		
Location	N	Cost of Hospitalization	Location	N	Cost of Hospitalization
Small intestine	85	749.21 ± 526.74	Entire intestine	187	654.20 ± 568.65
Ileocecal area	155	729.56 ± 485.70	Left large intestine	81	612.94 ± 498.67
Large intestine	68	697.67 ± 547.01	Rectum	52	559.66 ± 408.71
Without division according to location	308	725.02 ± 510.93	Without division according to coverage	320	628.40 ± 528.68

**Table 3 children-10-01112-t003:** The dependence of hospitalization costs for the studied group of children on disease activity (EUR).

	Crohn Disease		Ulcerative Collitis	
Activity Index PUCAI/PCDAI	N	Cost of Hospitalization	N	Cost of Hospitalization
Remission	65	552.37 ± 380.97	93	348.68 ± 336.59
Mild	161	654.59 ± 463.09	128	499.55 ± 375.53
Moderate	62	851.35 ± 397.50	81	965.25 ± 498.63
Severe	20	1457.94 ± 807.62	18	1474.01 ± 76.50

**Table 4 children-10-01112-t004:** The costs of outpatient treatment borne by the parents and carers of children with IBD (EUR).

	Number of Patients	Mean ± Standard Deviation	Median	Min. Value	Max. Value
Cost of the diet	61	84.99 ± 63.52	70.69	4.71	235.62
Cost of travel to treatment	61	23.28 ± 21.02	16.49	2.36	117.81
Cost of constant drugs	61	35.74 ± 22.76	35.34	2.36	117.81
Cost of periodic drugs	61	17.55 ± 18.52	11.78	4.71	70.69
Sum of IBD costs	61	161.57 ± 85.63	141.37	16.49	459.46
Sum of UC costs	36	146.98 ± 73.18	121.48	16.49	344.01
Sum of CD costs	25	182.56 ± 98.75	176.72	18.85	459.46

## Data Availability

The data presented in this study are available on request from the corresponding author.

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
