# Peer review of "Costs of Healthcare for Children with Inflammatory Bowel Diseases (IBD) in Poland"

_children, 2023, doi:10.3390/children10071112_

Round 1

Reviewer 1 Report

        - Use always abbreviations after you have defined them (see, for example, “inflammatory 50 bowel diseases”)

-        - Use oxford comma in the whole text

-        - Do not report the number of included patients in the Methods, but in the Result section

-        - “The documentation of 9 children”

Why 9?

-        - “Quantitative variables referring to the costs of treating patients for the needs of sta-76 tistical analyses were transcribed into qualitative variables considering the minimum, 77 maximum, and average values”

Mean or median? According to which statistical test?

-        - “Along the three year follow up, there was a downward trend in the average drug 104 costs (25%),”

Specify how the costs were calculated

-        - “Differences in hospitalization costs depending on age were also found among children with UC (p <0.01) (Table I).” “It was found” “Scientists from Germany” “The most important results were 20 determined with high” “During the observation, it was noticed that 23 the shortening of the hospitalization time did not significantly affect the 24 total costs, which remained at a stable level.”

Check the characters

-        - “p <0.00 and p <0.00”

Report at least one digit different from 0 after the “.”

-        - How is it possible that 65 patient with Crohn’s disease and 93 patients with UC were in remission at time of hospitalization (Table III)?

-        - Add the limitations to your study

Author Response

Thank you for the review. We have entered all the comments or explained any difficulties.

  1. Use always abbreviations after you have defined them (see, for example, “inflammatory 50 bowel diseases”. The comment has been taken into account.

  1. Use oxford comma in the whole text. The comment has been taken into account.

  1. Do not report the number of included patients in the Methods, but in the Result section. The comment has been taken into account.

  1. “The documentation of 9 children” Why 9? Record deleted. It was the result of a typographical error.

  1. “Quantitative variables referring to the costs of treating patients for the needs of sta-76 tistical analyses were transcribed into qualitative variables considering the minimum, 77 maximum, and average values”. Mean or median? According to which statistical test? In the text, the average age is given. Types of statistical tests are described in the section Materials and Methods: For statistical analysis of the collected material was used parametric and non-parametric dependency tests (Pearson and R Spearman X2) and variable independence tests (V-squared test, X2 test with Yates correction). In addition, the correlation strength between statistically significant variables was examined using appropriate correlation coefficients. Statistical analysis of the collected material was made using the following programs: Microsoft Office (MS Word, MS Excel, and MS Access) and Statistica 12, with the addition of the Medical Kit.

  1. “Along the three year follow up, there was a downward trend in the average drug 104 costs (25%)”. Specify how the costs were calculated. The costs were calculated on the basis of hospital documentation and settlements with the National Health Found. The information has been added to the text.

  1. “Differences in hospitalization costs depending on age were also found among children with UC (p <0.01) (Table I).” “It was found” “Scientists from Germany” “The most important results were 20 determined with high” “During the observation, it was noticed that 23 the shortening of the hospitalization time did not significantly affect the 24 total costs, which remained at a stable level.” Check the characters “p <0.00 and p <0.00”. The error has been corrected.

  1. How is it possible that 65 patient with Crohn’s disease and 93 patients with UC were in remission at time of hospitalization (Table III)? Patients have been diagnosed, which in the Polish system is associated with hospitalization.

  1. Add the limitations to your study This information has been added to the text.

Reviewer 2 Report

1. Kindly review the grammar and spelling errors in the following sections:

Lines 20-25, Lines 122-123, Lines 133-135, Lines 140, 147, 189, Lines 226-228

2. Please verify the references to ensure citation consistency.

Author Response

Thank you for the review. We have entered all the comments or explained any difficulties.

  1. Kindly review the grammar and spelling errors in the following sections:

Lines 20-25, Lines 122-123, Lines 133-135, Lines 140, 147, 189, Lines 226-228

The linguistic correction has been done.

  1. Please verify the references to ensure citation consistency.

Footnotes have been arranged in the order of citations.

Reviewer 3 Report

This is a retrospective analysis about costs of health care for children with inflammatory bowel diseases hospitalized in the Gastroenterology Ward of Silesia (Poland), along the three years follow up.
Please write the abstract and the article with the same font size.
You did not mention the exclusion criteria from the study. Please add this information.
Any study requires the informed consent of the individuals and considerations of ethical issues, and these must be approved by the Ethics Committee. You have not mentioned anything about this information.
You did not discuss the limits of the study.
It is not necessary to number the conclusions.
It is not necessary to specify that the source of the tables is your study, this aspect is obvious.
The article presents 28 references,. I recommend you to expand the discussions, so you will have a larger number of recent references.

Author Response

Thank you for the review. We have entered all the comments or explained any difficulties.

  1. Please write the abstract and the article with the same font size. An editorial error has been corrected.

  1. You did not mention the exclusion criteria from the study. Please add this information. The exclusion criteria concerned children in the questionnaire survey if the parents did not agree to its conduct. In the case of statistical data, all cases in the period were examined.

  1. Any study requires the informed consent of the individuals and considerations of ethical issues, and these must be approved by the Ethics Committee. You have not mentioned anything about this information. Such information is included in the text in the materials and method section: “Before the collection of data, the consent of the Bioethical Commission of the Medical University of Silesia no. KNW/0022/KB/308/15 was obtained”. In addition, parents gave written consent to participate in the study.

  1. You did not discuss the limits of the study. Strengths and limitations are the small number of children suffering from the discussed diseases and the limited number of publications in the field of health economics. This information has been added to the text.

  1. It is not necessary to number the conclusions. The comment has been taken into account.

  1. It is not necessary to specify that the source of the tables is your study, this aspect is obvious. The comment has been taken into account.

  1. The article presents 28 references,. I recommend you to expand the discussions, so you will have a larger number of recent references. In the available literature reports, there is a small number of publications defining the level of costs of diagnosis and treatment of adults with IBD and their complete lack in children and adolescents in the Polish population. Even fewer reports refer to the level of costs incurred by families and carers of children with IBD. The authors wanted the text to cover a narrow range of the costs of the disease, and such data is unavailable.

Reviewer 4 Report

The authors have performed a cost estimate over three years for paediatric patients hospitalised in Poland from a single centre in Poland. They found high costs, which tended to become more related to treatment and diagnostic tests rather than hospitalisation as time progressed. The grammar and layout of the manuscript are difficult to understand and extensive changes are needed. Also further clarification is needed on several aspects of the manuscript to help readers understand how the study was conducted. I have the following suggestions:

- Methods - more details are needed on how costs were calculated need to be provided – was the costs estimated or directly calculated? From whose perspective were costs take? Was it the patient or the health care payer? What sources were used to provide cost data? Was coding data used to determine the diagnosis of IBD? Similar details on how estimates of non-hospital costs need to be provided.

- Page 2, methods section - Details on the number of patients identified for the study should be omitted from the methods section and included in the results section. The methods should focus on the process that was used to identify patients.

- How were patients identified and selected for the study? Were all patients selected or how were patients determined to be eligible? Was there inclusion and exclusion criteria?

Statistical analysis – please provide details on which tests were used for assessing differences between groups e.g. t-test

- Results – consider using median results rather than averages for duration of stay as it is less affected by outliers

- For total costs, should these be per patient rather than absolute costs when comparing UC and CD data, given there are differences in the number of patients in each group? Consider revising.

- The discussion should compare the available literature to the current study and make comparisons to these, not just list the findings of other studies. Further discussions of the limitation of this paper are needed.

- Avoid numbering the conclusions

Minor points:

Page 3, line 98-99 “The age of children during the diagnosis of both diseases was, on average, two years lower than patients during hospitalization.” – this is difficult to understand, consider revising

Page 3, line 100 “diagnosing” should be “diagnosis”

Page 3, line 114 “analized” should be “analyzed”

Page 3, line 126 “location” should be “disease location”

Page 3, line 129 “the rarest” should be changed to “least frequent”

Page 6, line 239 “leaves” should be “leave”

Author Response

Thank you for the review. We have entered all the comments or explained any difficulties.

  1. Methods - more details are needed on how costs were calculated need to be provided – was the costs estimated or directly calculated? The costs were calculated directly. This information has been added to the text.

  1. From whose perspective were costs take? Was it the patient or the health care payer? What sources were used to provide cost data? The costs were calculated directly and concerned procedures administered in the hospital (analysis of settlements with the National Health Fund) and the costs incurred by parents. This information has been added to the text.

  1. Was coding data used to determine the diagnosis of IBD? The study group was selected according to the ICD10 diagnosis code: Crohn's disease has the code K50 and Ulcerative colitis has the code K51.

  1. Similar details on how estimates of non-hospital costs need to be provided. The study used a proprietary questionnaire addressed to parents and caregivers. Prior to data collection, consent was obtained from the Bioethics Committee of the Medical University of Silesia, no. KNW/0022/KB/308/15 for conducting research.

  1. Page 2, methods section - Details on the number of patients identified for the study should be omitted from the methods section and included in the results section. The methods should focus on the process that was used to identify patients. The comment has been taken into account.

  1. How were patients identified and selected for the study? Were all patients selected or how were patients determined to be eligible? Was there inclusion and exclusion criteria? The exclusion criteria concerned children in the questionnaire survey if the parents did not agree to its conduct. In the case of statistical data, all cases in the period were examined.

  1. Statistical analysis – please provide details on which tests were used for assessing differences between groups e.g. t-test. Types of statistical tests are described in the section Materials and Methods: For statistical analysis of the collected material was used parametric and non-parametric dependency tests (Pearson and R Spearman X2) and variable independence tests (V-squared test, X2 test with Yates correction). In addition, the correlation strength between statistically significant variables was examined using appropriate correlation coefficients. Statistical analysis of the collected material was made using the following programs: Microsoft Office (MS Word, MS Excel, and MS Access) and Statistica 12, with the addition of the Medical Kit.

  1. Results – consider using median results rather than averages for duration of stay as it is less affected by outliers. Indeed, using the median would be a better solution. Even so, using the average also gives a picture of reality. The use of the average has been additionally emphasized in the text so that the reader has no doubts.

  1. For total costs, should these be per patient rather than absolute costs when comparing UC and CD data, given there are differences in the number of patients in each group? Consider revising. I agree with the above comments. However, at this stage we would have to recalculate total costs. Please accept the text without this change. At the same time, thank you very much for this attention and we will try to conduct calculations in this way in the future.

  1. The discussion should compare the available literature to the current study and make comparisons to these, not just list the findings of other studies. Further discussions of the limitation of this paper are needed. Strengths and limitations are the small number of children suffering from the discussed diseases and the limited number of publications in the field of health economics. This information has been added to the text.

  1. Avoid numbering the conclusions. The comment has been taken into account.

  1. Minor points: Page 3, line 98-99 “The age of children during the diagnosis of both diseases was, on average, two years lower than patients during hospitalization.” – this is difficult to understand, consider revising. Children were hospitalized several times, so the age of hospitalization increased. It is a logical conclusion. The sentence has been removed.

Page 3, line 100 “diagnosing” should be “diagnosis”

Page 3, line 114 “analized” should be “analyzed”

Page 3, line 126 “location” should be “disease location”

Page 3, line 129 “the rarest” should be changed to “least frequent”

Page 6, line 239 “leaves” should be “leave” Changes have been implemented.

Reviewer 5 Report

In this study, authors investigated costs of health care for childred with IBD in Poland. Study is well written, and I have only few minor comments:

- Line 82-83 -  define in more detail what the survy consisted of and how the parents were approached and included in the investigation

- in Discussion, emphasize more the limitations of this study, as well as future perspectives and advices that could be derived from the results of this study

Author Response

Thank you for the review. We have entered all the comments or explained any difficulties.

  1. Line 82-83 -  define in more detail what the survy consisted of and how the parents were approached and included in the investigation. The comment has been taken into account in the materials and method section.

  1. In Discussion, emphasize more the limitations of this study, as well as future

perspectives and advices that could be derived from the results of this study. Strengths and limitations are the small number of children suffering from the discussed diseases and the limited number of publications in the field of health economics. This information has been added to the text.

Round 2

Reviewer 1 Report

The quality of the paper is insufficient.

Author Response

The text was resubmitted for linguistic correction. However, I would like to emphasis that already in the original version, „language editing by MDPI” was made by one of the co-authors: Dr. Wojciech Chlebowczyk.

Reviewer 2 Report

Accept in present form

Author Response

The text was resubmitted for linguistic correction. However, I would like to emphasis that already in the original version it was " language editing by MDPI" by one of the co-authors: Dr. Wojciech Chlebowczyk.

Reviewer 4 Report

The authors have made extensive changes to the manuscript as suggested. Some minor comments

Page 2, line 93-94 "Strengths and limitations..." this should be removed from the methods and added to the discussion section

Page 3, line 114 "The forms" should be changed to "Disease type" 

Author Response

Page 2, line 93-94 "Strengths and limitations..." this should be removed from the methods and added to the discussion section - The changes have been made to the text.

Page 3, line 114 "The forms" should be changed to "Disease type" - The chang has been made to the text.